# Effects of Icariin and Its Metabolites on GPCR Regulation and MK-801-Induced Schizophrenia-Like Behaviors in Mice

**DOI:** 10.3390/molecules28217300

**Published:** 2023-10-27

**Authors:** Su Hui Seong, Seo Hyun Kim, Jong Hoon Ryu, Jin-Woo Jeong, Hyun Ah Jung, Jae Sue Choi

**Affiliations:** 1Division of Natural Products Research, Honam National Institute of Biological Resources, Mokpo 58762, Republic of Korea; shseong@hnibr.re.kr (S.H.S.); jwjeong@hnibr.re.kr (J.-W.J.); 2Division of Research Management, Honam National Institute of Biological Resources, Mokpo 58762, Republic of Korea; kshgg0808@hnibr.re.kr; 3Department of Biomedical and Pharmaceutical Sciences, Kyung Hee University, Seoul 02447, Republic of Korea; jhryu63@khu.ac.kr; 4Department of Food Science and Human Nutrition, Jeonbuk National University, Jeonju 54896, Republic of Korea; 5Department of Food and Life Science, Pukyong National University, Busan 48513, Republic of Korea

**Keywords:** icariin, icariside II, GPCR, schizophrenia, MK-801

## Abstract

Icariin, a major bioactive compound found in the *Epimedium* genus, has been reported to exert protective effects against neurodegenerative disorders. In the current study, we aimed to investigate the regulatory effect of icariin and its active metabolites (icariside II and icaritin) against prime G-protein-coupled receptor targets, considering their association with neuronal disorders. Icariside II exhibited selective agonist activity towards the dopamine D3 receptor (D_3_R), with half-maximal effective concentrations of 13.29 μM. Additionally, they effectively inhibited the specific binding of radioligands to D_3_R. Molecular docking analysis revealed that icariside II potentially exerts its agonistic effect through hydrogen-bonding interaction with Asp110 of the D_3_R, accompanied by negative binding energy. Conversely, icaritin demonstrated selective antagonist effects on the muscarinic acetylcholine M2 receptor (M_2_R). Radioligand binding assay and molecular docking analysis identified icaritin as an orthosteric ligand for M_2_R. Furthermore, all three compounds, icariin and its two metabolites, successfully mitigated MK-801-induced schizophrenia-like symptoms, including deficits in prepulse inhibition and social interaction, in mice. In summary, these findings highlight the potential of icariin and its metabolites as promising lead structures for the discovery of new drugs targeting cognitive and neurodegenerative disorders.

## 1. Introduction

All fundamental processes within the central nervous system (CNS) rely on rapid and precise neuronal communication. Neurons form intricate circuits, ranging from simple di-synaptic feedback loops to extensive networks connecting various brain nuclei [1]. These networks can be modulated by G-protein coupled receptors (GPCRs), which control several key neuronal functions such as the release of neurotransmitters (dopamine, serotonin, acetylcholine, and adrenaline), neuronal excitability, and firing patterns of action potentials. Consequently, GPCRs have emerged as highly promising targets for CNS drug discovery in the treatment of schizophrenia, Parkinson’s disease (PD), Alzheimer’s disease (AD), and depressive disorders. Of the 826 human GPCRs, 165 non-olfactory GPCRs have been validated as drug targets [2]. Recent statistical data reveal that of all the Food and Drug Administration (FDA)-approved drugs, 527 drugs specifically target GPCRs (class A), with the highest proportion (26%) being drugs designed for CNS disorders [3].

In 2015, two new drugs, aripiprazole lauroxil (Aristada) and brexpiprazole (Rexulti), were approved by the FDA for the treatment of schizophrenia. These drugs act as partial agonists for dopamine D2/D3 and serotonin 1A receptors. In addition, the drug Lybalvi, a combination of samidorphan, an opioid receptor antagonist, and olanzapine, an atypical antipsychotic, was also approved by the FDA in 2021 [4]. A recent study showed that treatment with SK609, a selective dopamine D3 receptor (D_3_R) agonist and norepinephrine transporter (NET) inhibitor, alleviated motor symptoms and improved cognitive task performance in in vivo PD models without causing side effects associated with dopamine transporter activity [5]. However, most of the FDA-approved GPCR-targeted drugs are synthetic organic molecules, with only 16 derived from natural products [6]. Therefore, further investigation into lead compounds sourced from natural origins and the subsequent development of drugs utilizing these compounds is essential.

*Epimedium* (Berberidaceae), commonly known as the three branches-nine leaves grass, is one of the most well-known plants used in traditional Chinese medicine. The aerial parts of plants in this genus have been used in a popular botanical supplement formulated as a tonic and aphrodisiac in East Asia for over 2000 years [7]. Recently, several studies revealed that the genus *Epimedium* exerts protective effects against depression and neurodegenerative disorders (such as AD and PD) [8]. The active components of this genus are flavonoids, which are responsible for its biological actions in vitro and in vivo. Previous studies demonstrated that icariin is a major bioactive component of *Epimedium* and is considered a quality marker for *Epimedium*-based products [9,10]. Icariin is a class of prenylated flavonols with C4′-methoxyl, C3-rhamnosyl, C7-glucosyl, and C8-monoprenyl groups. Recent pharmacokinetic studies have shown that icariin is metabolized to icariside II, icaritin, and desmethylicaritin by the human microbiota [11]. Therefore, the main aim of this study was to discover target GPCRs for identifying the role and action mechanism of icariin and its metabolites in the prevention and care of neuronal disorders. In addition, in vivo experiments were conducted to assess the efficacy of icariin and its metabolites in alleviating MK-801-induced schizophrenia-like behaviors in mice.

## 2. Results

### 2.1. HPLC Analysis of EAEK

The HPLC chromatogram of the ethanolic extract of the aerial part of *Epimedium koreanum* (EAEK) is shown in Figure 1. The peaks were identified via comparison of the retention time with those of standard compounds **1**–**3**. As shown in Figure 1b, the main peak (18.92 min) of the EAEK was determined to be icariin (**1**). In addition, icariside II (**2**) was detected at 46.40 min, whereas icaritin (**3**) was not detected in the EAEK.

### 2.2. GPCR Target Screening of Icariin and Its Metabolites

To identify the major target GPCR of icariin and its metabolites, the modulatory effects of GPCR related to neuronal disorders were screened through functional assays. The receptors for dopamine, serotonin, vasopressin, and muscarinic acetylcholine (mACh) were selected as screening targets because they are associated with anxiety, schizophrenia, depression, and PD. The results of agonist/antagonist efficacy against various GPCRs are presented in Figure 2a. Functional assays for defining agonists revealed that icariside II showed significant agonist potential against D_3_R; 100 μM of icariside II inhibited NKH-477 (AC activator)-induced cAMP accumulation with an agonist effect of 100.95%, compared to the stimulatory effect of the control agonist (300 nM dopamine). Further evaluation showed a dose-dependent agonist effect of icariside II on D_3_R (up to 100 μM), yielding an EC_50_ value of 13.29 μM (Figure 2b). In contrast, icariin showed only a mild agonist effect of 28.35% stimulation at 100 μM. However, icaritin, which is an aglycone of icariin and icariside II, showed no agonist activity against any of the GPCRs at 100 μM. Thus, it was discovered that glycosides with a sugar linked to the C3- or C7-position are more efficient than icaritin itself in the D_3_R agonist action. Moreover, functional assays for defining antagonists revealed that icariside II at a concentration of 100 μM exerted a strong antagonist effect (42.0%) on vasopressin 1A receptor (V_1A_R). In addition, icaritin showed a moderate antagonist effect against M4 mACh receptor (M_4_R), with 32.4% inhibition of the control agonist (100 nM acetylcholine (ACh)) response. In the agonist effect assay on M_4_R, icariin demonstrated the opposite effect to that of icariside II, which has an agonist effect. These findings are taken to imply that icariin may work as an inverse agonist. We further evaluated the selectivity of icaritin to mACh receptors. Icaritin had higher selectivity for M_2_R than for M_4_R and had no effect on M_1_R, M_3_R, and M_5_R (Figure 2c). Icaritin at a concentration of 100 μM exerted an antagonist effect against M_2_R by 54.55% through competition with the control agonist (300 nM ACh). Unfortunately, icariin and its metabolites did not exhibit any agonist/antagonist activity against dopamine D4 receptor (D_4_R), serotonin 1A receptor (5HT1AR), and calcitonin gene-related peptide receptor (CGRP) at a concentration of 100 μM.

### 2.3. Agonist Effect of Icariside II on D_3_R

In the functional assay, icariside II showed a significant and selective agonist effect against D_3_R (EC_50_ = 13.29 μM). Thus, the binding affinity of icariside II was measured via a radioligand binding assay using [^3^H]methyl-spiperone on the recombinant CHO-D_3_R cell membrane homogenates. As shown in Figure 3a and Appendix A, icariside II inhibited radioligand-specific binding in a dose-dependent manner with an IC_50_ value of 19.46 μM. In particular, radioligand binding to the target binding site was almost completely blocked by 50 μM icariside II. To observe the binding sites between D_3_R-icariside II, in silico molecular docking analysis was performed, and two hit poses were generated (Figure 3c). Several studies have demonstrated that the conserved Asp110 residue in transmembrane (TM)-3 is critical for D_3_R activation by an agonist. As shown in Figure 3c and Appendix A, the C-7 hydroxyl moiety of icariside II tightly interacted with Asp110 via hydrogen bonding in pose-1 and pose-2. In addition, the methyl group (C-6′) at the rhamnosyl moiety and prenyl moiety of icariside II formed stable hydrophobic interactions with D_3_R residues. Taken together, the results of the radioligand binding assay and molecular docking study clearly demonstrated that the hydrogen-bonding interaction with conserved aspartic acid residues as well as hydrophobic interactions with nonpolar residues are potential mechanisms by which icariside II binds to D_3_R to exert its agonist effect.

### 2.4. Antagonist Effect of Icaritin on M_2_R

In the functional assay, icaritin showed a selective partial antagonist effect against M_2_R. Thus, the binding affinity of icaritin was measured by a radioligand binding assay using [^3^H]AF-DX 384 on the CHO-M_2_R cell membrane homogenates. As shown in Figure 3b, 150 μM icaritin blocked radioligand binding to the target binding site by 36%. Molecular docking analysis was performed to observe the M_2_R–icaritin binding characteristics, and two distinguished hit poses were observed (Figure 3d). The importance of the conserved Asp103 residue in TM-3 and allosteric site residues (three Tyr residues) for ligand binding was demonstrated in a previous study [12,13]. In the pose-1, the aromatic B-ring of icaritin formed an electrostatic interaction (pi-anion) with Asp103 as well as a π–π interaction with Trp400 and Tyr403. In addition, three hydrogen-bonding interactions were observed between the hydroxyl/ketone moieties of icaritin and M_2_R residues (Ala191, Tyr104, and Tyr403). Furthermore, the prenyl moiety of icaritin formed stable hydrophobic interactions (π–alkyl/alkyl) with Val111, Trp155, and Ala194. In the pose-2, on the other hand, icaritin formed hydrogen-bonding interactions with Ile178, Asn419, and allosteric site residues (Tyr403 and Tyr104). In addition, the A-ring and prenyl moiety of icaritin formed stable pi-interactions with allosteric site residue, Tyr426. Therefore, the role of icaritin as a selective M_2_R antagonist was corroborated by the radioligand binding assay, antagonist effect assay, and molecular docking analysis.

### 2.5. hMAO and Lipid Peroxidation-Inhibitory Activity of Icariin and Its Metabolites

Previous studies have shown that increased MAO activity and levels of iron in the brain are key pathogenic factors in neuronal disorders [14]. Iron induces oxidative stress and promotes the breakdown of lipid peroxides, resulting in extremely cytotoxic free radicals [15]. Thus, we investigated the inhibitory effect of icariin and its derivatives on MAO and lipid peroxidation. Our results showed that icaritin and icariside II exerted significant inhibitory effects against lipid peroxidation, with IC_50_ values of 81.33 and 70.38 μM, respectively (Table 1). However, icariin showed weak inhibitory activity, with an IC_50_ value of 301.03 μM. In the case of MAO-A inhibition, icariin showed weak inhibitory activity (IC_50_ = 421.74 μM), whereas the other compounds showed no significant effect. In case of MAO-B inhibition, all three compounds showed no activity under the tested concentrations.

### 2.6. Effects of Icariin Derivatives on the Loss of Sensorimotor Gating Function Induced by MK-801 in the Acoustic Startle Response Task

A prepulse inhibition (PPI) test is a widely used paradigm for assessing sensorimotor gating in humans and rodents. The PPI of the startle response (SR) occurs when a mild prepulse stimulus is provided 30–500 ms before the startling stimulus. Patients with psychiatric diseases like schizophrenia have been observed to have reduced PPI, which is assumed to represent sensorimotor gating failure [16]. Thus, the acoustic SR task (ASRT) was used to evaluate the effects of icariin and its derivatives on sensorimotor gating function in MK-801-treated mice. As shown in Figure 4a,b, the administration of 0.2 kg/mg MK-801 alone (i.p.) significantly increased the acoustic startle amplitude at 120 dB and decreased the prepulse inhibition (PPI) at +3, +6, +12, and +16 dB above the background noise (70 dB), which explains the MK-801-induced sensorimotor gating deficits. In contrast, icariin (3 and 10 mg/kg, p.o.) and the positive control (1 mg/kg aripiprazole, i.p.) significantly reversed MK-801-induced PPI deficits at +12 and +16 dB above the background noise (Figure 4b), whereas icariin metabolites (icariside II and icaritin) showed no significant effects under the tested concentrations (1, 3, and 10 mg/kg, p.o.).

### 2.7. Effects of Icariin Derivatives on Social Ability and Interaction in the Social Novelty Preference Test

To assess the effects of icariin and its derivatives on social ability and interaction in MK-801-treated mice, we performed the social novelty preference test (SNPT). The primary principle of this experiment is based on a subject mouse’s free choice to spend time in any of three box compartments throughout two experimental sessions, involving indirect interaction with one or two unfamiliar mice [17]. As shown in Figure 5b,d,f, the administration of MK-801 alone (0.2 mg/kg, i.p.) considerably reduced the amount of time spent in the chamber with the unknown mouse when compared with that in the control group (Con), indicating that the MK-801-treated mice failed to show any preference for the unfamiliar mouse. In contrast, icariin (3 and 10 mg/kg, p.o.), icariside II (1, 3, and 10 mg/kg, p.o.), and icaritin (3 and 10 mg/kg, p.o.) significantly reversed MK-801-induced social interaction deficits, as observed with the positive control, aripiprazole (1 mg/kg, i.p.).

## 3. Discussion

The human gut microbiota, consisting of symbiotic bacteria, has a significant impact on the metabolism of bioactive nutraceuticals derived from the diet, generating a diverse array of metabolites [18]. Hence, it is crucial to investigate in vitro assays using microbial metabolites that effectively interact with biochemical receptors in vivo. In a previous study, the metabolism of icariin by human intestinal bacteria was examined to determine if this compound serves as the active phytochemical responsible for its biological effects within the body. Notably, icariin was rapidly metabolized (within 1 h) into icariside II by the human intestinal microflora [11]. In addition, Xu and coworkers [19] reported the complete metabolic profile of icariin in rats and suggested that icariside II and icaritin are the most abundant metabolites of icariin in rat feces after oral administration. Icariside II has a structure in which one glucose moiety is eliminated from the C7 position of icariin through deglycosylation, whereas icaritin has a non-saccharide structure with rhamnose and glucose moieties detached from the C3 and C7 positions of icariin, respectively. Icariside II and icaritin have demonstrated potent anticancer, anti-inflammatory, anti-cholinesterase, and anti-phosphodiesterase-5 activities [20,21,22]. In addition, icariside II showed neuroprotective activity via modulation of beta-amyloid production and various pathways, including the MAPK, TLR4/MyD88/NF-kB, and Keap1-Nrf2 pathways in preclinical studies [23,24]. Recent research has also indicated that icaritin mitigates Parkinson’s disease (PD) by alleviating mitochondrial damage and oxidative stress in a 1-methyl-4-phenyl-1,2,3,6-tetrahydropyridine (MPTP)-induced mouse model of PD [25]. Previous research has found that icariside II effectively attenuated blood-brain barrier (BBB) disruption after cerebral ischemia/reperfusion injury in rats [26]. In addition, Feng and coworkers demonstrated that icariside II and icaritin, which are major metabolites of icariin, could penetrate the blood-brain barrier (BBB) readily [27]. Nevertheless, the direct mechanisms through which icariin and its metabolites act on neuronal GPCRs remain unreported.

Dopamine, the most abundant and functionally significant neurotransmitter in the CNS, is synthesized from L-tyrosine in the dopaminergic neurons and released to stimulate GPCRs, thereby affecting cognition, memory, movement, reward, and emotion. In the brain, dopamine acts through five different dopaminergic receptors, which are classified into two subfamilies, D1-like (D_1_R and D_5_R) and D2-like (D_2_R-D_4_R) receptors, based on pharmacological and physiological properties [28]. The distinct distribution of D_3_R in limbic brain regions, which are crucial in the regulation of reward, cognitive, and motivation functions, has sparked interest in D_3_R as a possible target for CNS drug discovery [29]. D_3_R agonists are considered to be effective in the treatment of PD by boosting the dopamine content, reducing α-synuclein accumulation, enhancing brain-derived neurotrophic factor secretion, promoting neurogenesis, and interacting with D_1_R to reduce the motor symptoms of PD [30]. In case of D_2_R/D_3_R partial agonists, they are considered new atypical anti-psychotic drugs for the treatment of schizophrenia. A high affinity for D_3_R in combination with the D_2_R affinity of cariprazine (Vraylar^®^) may offer the potential for augmented effects on the cognitive deficits and negative symptoms of schizophrenia [31]. In the present study, icariside II, the primary metabolite of icariin, showed selective agonist effects against D_3_R in the functional cAMP assay. In addition, the radioligand binding assay demonstrated that icariside II could sufficiently bind to the active site of D_3_R at the active doses of this compound observed in the functional cAMP assay. Furthermore, molecular docking simulation confirmed that icariside II formed a stable and strong hydrogen bond with Asp110, a conserved orthosteric binding site (OBS) residue of D_3_R. These in vitro and in silico results propose that icariside II could enhance dopaminergic function by acting as a D_3_R agonist that not only decreases cAMP formation but also regulates the Akt/glycogen synthase kinase-3β (GSK-3β) signaling pathway [32]. This activation of the dopaminergic signaling pathway by D_3_R agonist action can effectively prevent motor symptoms of PD but has caused side effects related to cognitive function. However, recent studies have reported that multi-targeted ligands, such as SK609 targeting D_3_R and NET, can reduce these side effects [5].

In this study, icariside II not only showed the D_3_R agonist effect but also showed a V_1A_R antagonist effect. Vasopressin (AVP), a neuropeptide with three receptors: V1A, V1B, and V2, regulates amygdaloid activity in response to threatening stimuli as well as anxious mood. Since V_1A_R-expressing regions modulate anxiety and fear, studies on rodents have mainly examined how V_1A_R affects anxious behavior [33]. Recently, anthraquinone derived from Cassia seed, which is a dual D_3_R/V_1A_R agonist, significantly alleviated memory impairment and neuronal damage in the transient forebrain ischemia animal model [34]. Taken together, it is believed that icariside II could improve motor symptoms seen in neurodegenerative disorders through D_3_R agonist action and alleviate anxious behavior and memory performance through the V_1A_R antagonist action. However, additional mechanism studies and in vivo validation should be conducted.

By contrast, icaritin had a marked antagonist effect on mACh receptors with no modulatory effect on dopamine, serotonin, and vasopressin receptors. Similar to dopamine receptors, the mACh receptor family also consists of five distinct subtypes (M_1_R to M_5_R) that share 64–82% sequence identity and is classified into two classes: receptors that bind to the G_q/11_ protein (M_1_R, M_3_R, and M_5_R) and receptors that bind to the G_i/o_ protein (M_2_R and M_4_R) [35]. The mACh receptors are the targets of approved and developing drugs for various debilitating conditions, including AD, schizophrenia, cardiovascular disease, incontinence, and motion sickness [36]. Because of the high conservation of the OBS, developing a highly subtype-selective agonist/antagonist for mACh receptors has been challenging and largely unsuccessful. M_2_R plays a role as a presynaptic auto-receptor and controls the release of ACh. Presynaptic M_2_R inhibits ACh release when ACh levels increase rapidly, such as following injection of an acetylcholinesterase inhibitor; however, this constraint can be reversed by concurrent application of an M_2_R antagonist [37]. Thus, selective M_2_R antagonism in the brain, resulting in increased cholinergic transmission, has been proposed as an approach to improve cholinergic function in AD and related disorders. In the present study, icaritin showed antagonistic effects only on M_2_R and M_4_R, among the five mACh receptor subtypes. In particular, icaritin showed higher (1.58 times) selectivity for M_2_R than for M_4_R. In addition, the radioligand binding assay and molecular docking analysis revealed that icaritin is an orthosteric ligand, binding to active site residues, including Asp103 (a conserved OBS residue), Tyr104, Ala191, and Ala194. Previous studies showed that mutations of Asp103 and Tyr104 reduced the ligand-binding affinity by more than 10-fold [12]. As a result, this study showed that icaritin has the potential to be employed as a medication to improve cognitive performance by functioning as an M_2_R antagonist that can control Ach release.

Schizophrenia is a chronic, debilitating psychiatric disorder distinguished by positive (delusion, hallucinatory behavior, suspiciousness, and paranoia) and negative symptoms (flat affect, amotivation, alogia, and a sociality) as well as cognitive impairments (disturbed mental processing and disturbances in working memory). Among the various hypotheses related to schizophrenia, many studies have shown that *N*-methyl-D-aspartate (NMDA) hypofunction causes symptoms of the disease. NMDA receptor antagonists can elicit schizophrenia-like behaviors in humans and animals and are commonly employed in the development of anti-psychiatric medication [38]. Recently, Pan et al., reported that icariin is effective in ameliorating schizophrenia-like symptoms including anxiety, recognition memory deficits, and weakened motor coordination in an MK-801-induced rat model [39]. They also discovered that icariin may have an anti-schizophrenia effect via controlling the miR-144-3p/ATP1B2/mTOR signaling pathway. However, the effects of icariin on PPI and social deficits, both of which are representative symptoms of schizophrenia, have not been reported; furthermore, there are no reports of the efficacy of major icariin derivatives in treating schizophrenia-like behaviors. In the current investigation, acute treatment with MK-801 resulted in social recognition impairments and sensorimotor gating function deficits in mice, which was consistent with earlier in vivo observations [40]. The PPI deficit is one of the most characterized and highly reproducible symptoms in patients with schizophrenia. The Crawley’s three-chambered social task assesses social motivation and affiliation as well as social memory and novelty [41]. Aripiprazole, a D_2_R/D_3_R partial agonist, was used as a positive control in the behavior tests, and positive and negative symptoms induced by MK-801 were significantly alleviated even at a low concentration of 1 mg/kg. Icariin, a major component of *Epimedium* sp., significantly reversed MK-801-induced PPI deficits and social interaction impairments in mice. Additionally, the MK-801-induced impairment of social interaction and social preference was reversed by icariin and its active metabolites, icariside II and icaritin. The brain is highly vulnerable to oxidative damage because its membranes contain a significant amount of poly-saturated fatty acids with high peroxide values [42]. In this study, icariside II and icaritin showed potent inhibitory activity against lipid peroxidation in the ferrous-treated rat brain homogenates at concentrations below 100 μM.

## 4. Materials and Methods

### 4.1. Chemicals and Reagents

Human monoamine oxidase (hMAO) isozymes, icariin, icariside II, icaritin, sodium dodecyl sulfate, formic acid, 1,1,3,3-tetramethoxypropane, R-(–)-deprenyl HCl, and Trolox were purchased from Sigma-Aldrich (St. Louis, MO, USA). Thiobarbituric acid (TBA) was purchased from Tokyo Chemical Industry (Tokyo, Japan). All solvents used for component analysis were of HPLC grade or higher.

### 4.2. Plant Material

*E. koreanum* was collected from Mt. Nochu (Jeongseon-gun, Republic of Korea) in May 2019. The plant was authenticated by TJ Lee, the Director of the Hantaek Botanical Garden, Korea (voucher no. KHG2016-01-0015). EAEK was obtained from the KIST natural products library, Republic of Korea (registration no. BE0594H1).

### 4.3. HPLC Analysis of Icariin and Its Metabolites in EAEK

Metabolite analyses were performed using the ThermoScientific™ Ultimate™ 3000 UHPLC system with a diode array detector and Chromeleon Software 7.3 (Thermo Scientific, Pittsburgh, PA, USA). Separation was achieved using a Phenomenex C_18_ LUNA column (15 cm, 5 μm, 150 × 4.6 mm) with constant column temperature (25 °C). The mobile phase consisted of 0.1% HCOOH (A) and 0.1% HCOOH in MeCN (B) in the following gradient system: 0–8 min, B 17–27%; 8–32 min, B 27%; 32–60 min, B 27–85%; 60–60.1 min, B 85–100%; 60.1–65 min, B 100%. The other settings were as follows: injection volume, 5 μL; flow rate, 1.0 mL/min; and wavelength, 273 nm.

### 4.4. FRET-Based cAMP Assay

The transfected CHO-K1-GPCR cells, suspended in Hanks’ balanced salt solution (HBSS) buffer containing 20 mM HEPES (pH 7.4) and 0.5 mM IBMX, were seeded in well-plates at a density of 10^4^ cells/well. Subsequently, the test compound (icariin, icariside II, and icaritin), positive control, or buffer were added to each well at a final concentration (f.c.) of 100 μM each. Thereafter, NKH-477 was added to each well (together with the reference agonist in the case of antagonist assays). After 10 min incubation at 37 °C, the cells were lysed and D_2_-labeled cAMP (acceptor) and antibody with Eu^3+^ cryptate-labeled antibody (donor) were added. After 1 h incubation at 25 °C, fluorescence was measured at two different wavelengths (λ_em_ = 620 nm (A) and 665 nm (B)/λ_ex_ = 337 nm) using a microplate reader (EnVision, Perkin Elmer, Waltham, MA, USA). The cAMP level was calculated by dividing the signal observed at A by that observed at B. Details of the experimental conditions are provided in Appendix A.

### 4.5. Measurement of Intracellular Ca^2+^ Level

The *GPCR*-transfected cells suspended in Dulbecco’s modified Eagle medium or HBSS buffer were seeded in well-plates at a density of 2–3 × 10^4^ cells/well. A fluorescent probe (Fluo-4 and Fluo-8) mixed with probenecid in HBSSM buffer containing 20 mM HEPES (pH 7.4) was added to each well. After 1 h at 37 °C, the test compound (100 μM f.c. of icariin, icariside II, and icaritin), positive control, or buffer were added to each well. In the case of the antagonist assay, the stimulant was added to each well after 5 min. The fluorescence intensity was measured using a microplate reader (CellLux, PerkinElmer, Waltham, MA, USA). A detailed description of the experimental conditions is provided in Appendix A.

### 4.6. Radioligand Binding Assay

Human D_3_R and muscarinic acetylcholine M2 receptor (M_2_R) binding was evaluated using membrane homogenates of the transfected CHO-K1 cells, which were prepared as described in a previous study [43]. The membrane homogenates (8 μg protein) were incubated in the reaction buffer, with the radioligand (3 nM [^3^H] methyl-spiperone for D_3_R and 2 nM [^3^H]AF-DX 384 for M_2_R) and test compound (f.c.: 12.5–100 μM for icariside II and 100–150 μM for icaritin) at 22 °C for 1 h. Next, 10 μM (f.c.) (+)-butaclamol (for D_3_R) and 1 μM (f.c.) atropine (for M_2_R) were used to detect nonspecific binding in the respective assays. The binding affinity of a test compound was calculated as the inhibition (%) of control (radioligand)-specific binding.

### 4.7. In Silico Docking Simulation

X-ray crystallographic structures of proteins and three-dimensional structures of test compounds were obtained from the Protein Data Bank (ID: 3pbl for *h*D_3_R and 3uon for *h*M_2_R) [12,28] and PubChem (ID: 5318980 for icaritin and 5488822 for icariside II), respectively. For each ligand–protein complex, 15 docking postures were created with the same parameters (autogrid and autodock) using AutoDock 4.2.6 [44]. A grid box sized 80 × 90 × 80 with a spacing of 0.375 Å was used to cover the favorable target binding sites. The grid centers were 0.86, −15.073, and 9.582 for 3pbl and 6.663, 0.524, and −2.976 for 3uon. The posture in the most populated cluster with the lowest binding energy was selected as the ultimate docking outcome. Results were investigated using Discovery Studio (v17.2) and UCSF Chimera (v1.13.1, Accelrys, San Diego, CA, USA).

### 4.8. In Vitro hMAO-Inhibitory Assay

A homogeneous luminescent assay using the MAO-Glo™ assay kit (Promega, Madison, WI, USA) was used to evaluate the *h*MAO-inhibitory activity of icariin and its metabolites according to the manufacturer’s protocol. R-(–)-Deprenyl HCl (f.c.: 0.8–20 μM for MAO-A, 0.04–5 μM for MAO-B) was used as a reference compound.

### 4.9. Lipid Peroxidation Assay

The animal protocol was approved by the Pusan National University Institutional Animal Care and Use Committee (PNU-IACUC; Approval Number: PNU-2017-1428; Date: 18 July 2017). Whole brain homogenates from male Sprague-Dawley rats (6 months old) were prepared according to procedures described in previous studies [42,45]. The test compound (f.c.: 50–400 μM) was mixed with the supernatant of the homogenate, 250 μM ferric sulfate, and distilled water (5:10:3:12 ratio [*v*/*v*]). After 1 h of incubation at 37 °C, the stop solution was added to the reaction mixture [45]. After incubation in boiling water for 1 h, the reaction mixture was extracted with normal butanol and centrifuged (1300× *g* for 10 min). The MDA-TBA adduct content was calculated by measuring the absorbance of the organic fraction at 532 nm using a Gemini XPS (Molecular Devices, Sunnyvale, CA, USA). 1,1,3,3-Tetramethoxypropane (standard curve: 1–100 μM) and Trolox (f.c.: 15–75 μM) were used as the TBARS standard and positive control, respectively.

### 4.10. Animals

ICR male mice (26–28 g, five weeks old) were obtained from Orient Co., Ltd., a branch of Charles River Laboratories (Seongnam-si, Gyeonggi-do, Republic of Korea). The mice were housed in the University Animal Care Unit for one week prior to the experiment. Each cage housed 10 mice, and all mice had unlimited access to water and food. Animals were housed in a humidity (60 ± 10%) and temperature (23 ± 1 °C)-controlled environment with 12 h light and dark cycles. The treatment and care of animals were carried out in accordance with the Care and Use of Laboratory Animals (National Institutes of Health publication No. 85–23, revised 1985). The Institutional Animal Care and Use Committee of Kyung Hee University approved the mouse experimental protocols (approval number: KHUASP-22-530).

### 4.11. Drug Administration

Icariin, icariside II, and icaritin were dissolved in 10% Tween 80 solution. MK-801 and aripiprazole (positive control) were dissolved in 0.9% saline solution and distilled water with a few drops of Tween 80, respectively. MK-801 (0.2 mg/kg, i.p.) or aripiprazole (1 mg/kg, i.p.) was administered to the mice 30 min prior to each task. Icariin (1, 3, and 10 mg/kg, p.o.), icariside II (1, 3, and 10 mg/kg, p.o.), icaritin (1, 3, and 10 mg/kg, p.o.), or the vehicle were administered 1 h before each task.

### 4.12. Acoustic Startle Response Task (ASRT)

The ASRT was conducted using SR-LAB startle chambers (San Diego Instruments, San Diego, CA, USA), consisting of a box that can lightly confine the mice, a system that can generate sound stimuli regularly, equipment that can measure animal movements, a program that can digitize the measured stimuli, and a source and speaker that can provide sound. The ASRT sessions included three trials: non-stimulus (PPI-1), startle (pulse alone, PPI-2), and prepulse (prepulse + pulse, PPI-3). The PPI-1 trial consisted of background noise only. The PPI-2 trial consisted of constant noise (80−120 dB, five sections) for 40 ms. The PPI-3 trial consisted of a noise prepulse (73, 76, 82, and 86 dB, which are 3, 6, 12, and 16 dB above the background noise, respectively) for 20 ms, a delay for 80 ms, and a 120 dB startle pulse for 40 ms. The percentage score for PPI was calculated as follows: % PPI = [(PPI-2 – PPI-3)/(PPI-2)] × 100(%).

### 4.13. Social Novelty Preference Task (SNPT)

Three chambers in a rectangular Plexiglas box (w × d × h: 94 × 28 × 30 cm) were used for the SNPT. The left, right, and central chambers were all joined together by openings (w × d: 6 × 28 cm). On the center of the left and right chambers, an iron muntin cylinder lockup (h × d: 11 × 12 cm) was installed. A test mouse was allowed to interact with another mouse that was trapped in the cylinder lockup. A test mouse was placed in the middle chamber during the acclimatization trial (trial-1, 10 min) and was allowed to freely explore all three compartments through the apertures. A mouse that was younger than the test mouse was placed in a lockup in the first sociability experiment (trial-2) following trial-1, while the other lockup remained. The test mouse was taken to the center compartment and allowed to wander around. The second sociability trial (trial-3) was performed immediately after trial-2. A new mouse that was younger than the test mouse was put in the empty cell during trial-3 (10 min), and the test mouse was put into an empty lockup and allowed to explore. The video-based EthoVision system (Noldus) recorded all of the experiments, and the exploration time for each animal was analyzed by a researcher who did not participate in the task and was unaware of the treatment conditions.

### 4.14. Statistical Analysis

In vitro experimental data are expressed as the mean ± standard deviation of three independent in vitro experiments. In vivo experimental data were reported as the means ± standard error of the mean (SEM). Data from the exploration time (%) of SNPT and startle amplitude and the PPI (%) of ASRT were analyzed by two-way analysis of variance (ANOVA) followed by Bonferroni’s post hoc test. Statistical significance was set at *p* < 0.05, *p* < 0.001, and *p* < 0.0001.

## 5. Conclusions

Icariside II and icaritin, the active metabolites of icariin, have demonstrated efficacy in regulating key neuronal GPCRs, specifically D_3_R, V_1A_R, and M_2_R, whereas icariin itself exerts a weak D_3_R agonist effect, as shown in in vitro functional assays. Icariside II could enhance the dopaminergic function by acting as a selective D_3_R agonist that not only inhibits cAMP formation but also regulates the Akt/GSK-3β signaling pathway. In addition, icariside II could alleviate anxious behavior and memory performance through the V_1A_R antagonist action. Furthermore, icaritin could increase the availability of acetylcholine by acting as an M_2_R antagonist, thereby improving learning and memory performance. As D_3_R, V_1A_R, and M_2_R are closely related to cognitive and psychiatric disorders, the direct agonist/antagonist effects of icariside II and icaritin on these receptors, coupled with their dose-dependent lipid peroxidation-inhibitory effects, may elucidate the neuroprotective mechanism underlying the effects of *Epimedium* sp. and icariin. In addition, icariin and its metabolites have shown significant potential in reversing the MK-801-induced impairment of social interaction in mice. Our novel findings suggest that icariin and its metabolites might be promising lead structures in CNS drug development aimed at the prevention and treatment of neuronal disorders. However, further investigations are required to delve into the underlying mechanisms and ascertain the efficacy of icariin and its metabolites against various psychiatric disorders.

## Figures and Tables

**Figure 1 molecules-28-07300-f001:**
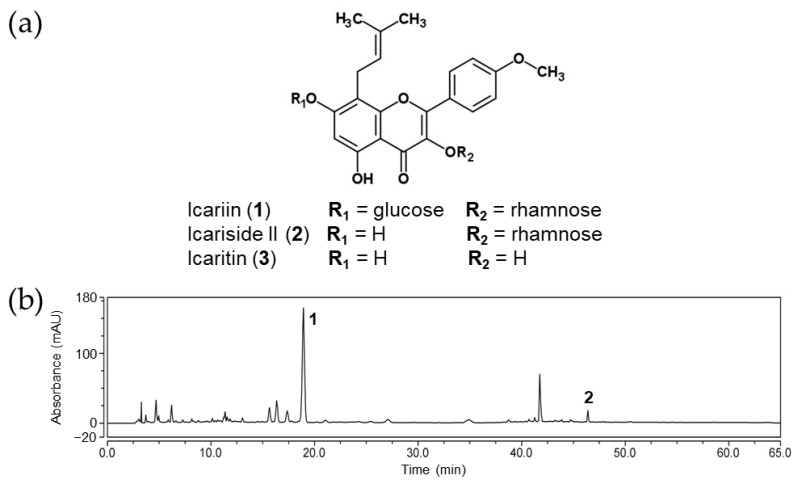
(**a**) Chemical structures of icariin and its metabolites. (**b**) HPLC chromatogram of EAEK (the peaks marked with **1** and **2** were icariin and icariside II, respectively).

**Figure 2 molecules-28-07300-f002:**
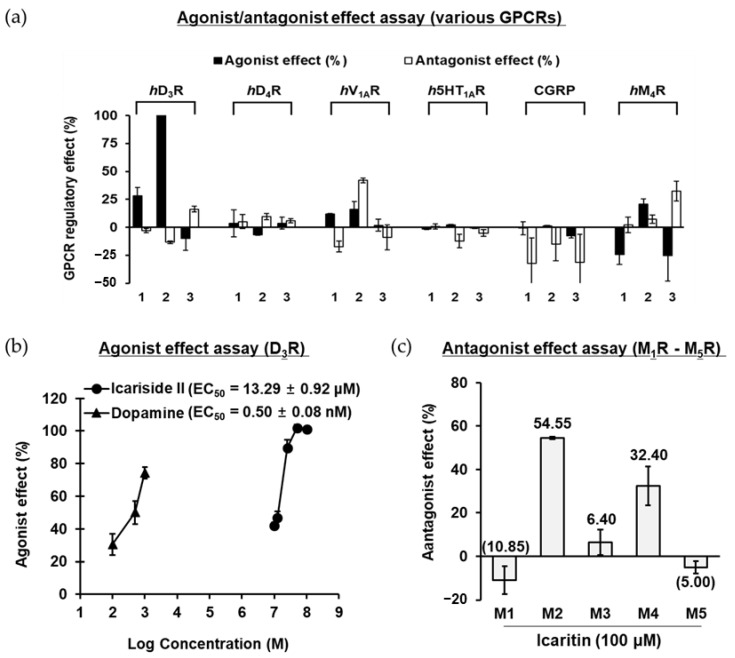
(**a**) Agonist and antagonist effect of compounds (**1**−**3**: 100 μM of icariin, icariside II, and icaritin) on various GPCRs. (**b**) Dose-dependent agonist effect of icariside II on *h*D_3_R. (**c**) Antagonist effect of icaritin on human muscarinic acetylcholine M_1_–M_5_ receptors at 100 μM.

**Figure 3 molecules-28-07300-f003:**
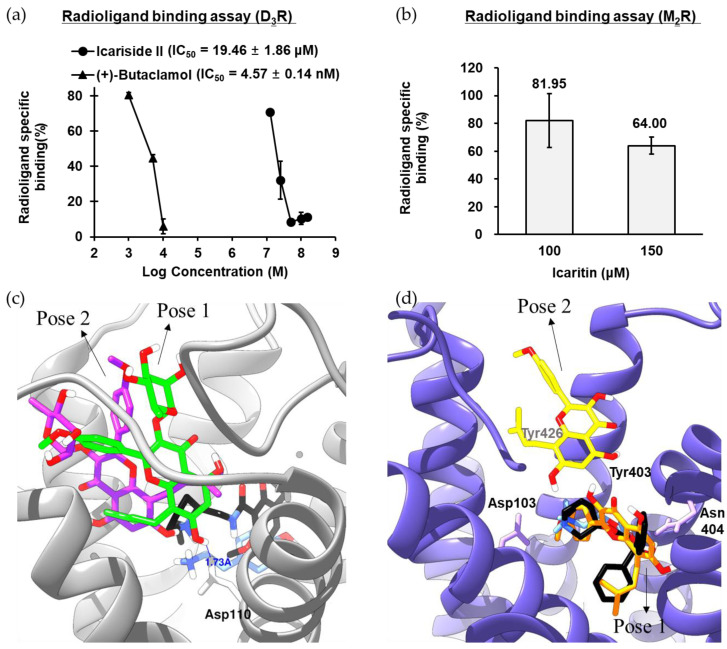
(**a**) D_3_R binding affinity of icariside II in [^3^H]methyl-spiperone competition binding assays to membrane homogenates from CHO cells stably expressing *h*D_3_R. (**b**) M_2_R binding affinity of icaritin in [^3^H]AF-DX 384 competition binding assays to membrane homogenates from CHO cells stably expressing M_2_R. (**c**) Molecular interactions between icariside II and specific residues of *h*D_3_R. Icariside II in pose-1 and pose-2 is presented as green and purple sticks, respectively. Dopamine and eticlopride are presented as blue and black sticks, respectively. The white represents hydrogen, and the red represents oxygen atoms. *h*D_3_R is presented as a gray ribbon representation. (**d**) Molecular interactions between icaritin and specific residues of *h*M_2_R. Icaritin in pose-1 and pose-2 is presented as orange and yellow sticks, respectively. Acetylcholine and QNB are presented as blue and black sticks, respectively. The white represents hydrogen, and the red represents oxygen atoms. *h*M_2_R is presented as a purple ribbon representation.

**Figure 4 molecules-28-07300-f004:**
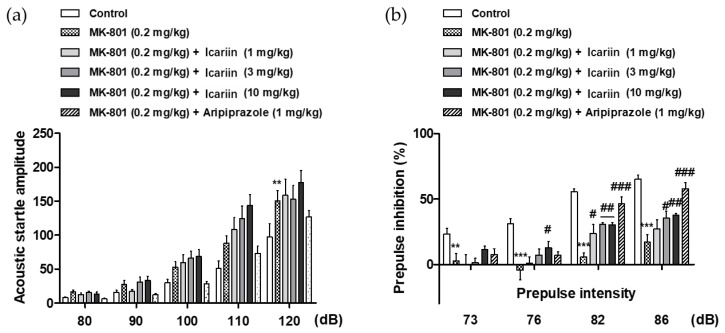
(**a**) Effects of icariin on the MK-801-induced sensorimotor gating function deficits in the acoustic startle response test in mice. The acoustic startle amplitude at each pulse intensity and (**b**) the percentage of prepulse inhibition are presented. The data represent the means ± S.E.M. (*n* = 9–10 per group) (** *p* < 0.001 and *** *p* < 0.0001 versus control, (^#^
*p* < 0.05, ^##^
*p* < 0.001 and ^###^
*p* < 0.0001 versus vehicle group).

**Figure 5 molecules-28-07300-f005:**
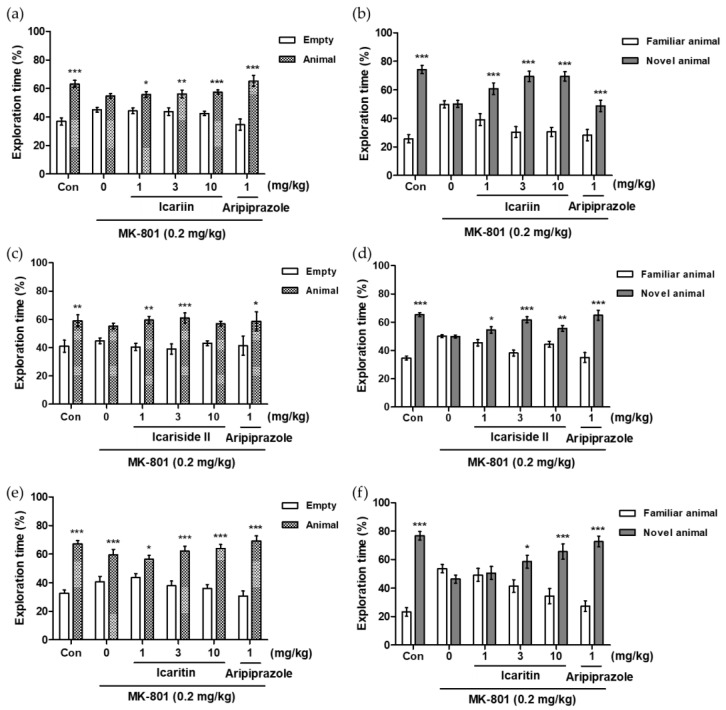
Effects of icariin derivatives on the MK-801-induced social interaction deficits in the social novelty preference test in mice. The percentages of exploration time spent in each chamber are presented (**a**,**c**,**e**): training trial to measure social recognition; (**b**,**d**,**f**): examining trial to measure social interaction). The data represent the means ± S.E.M. (*n* = 8–10 per group) (* *p* < 0.05, ** *p* < 0.001, *** *p* < 0.0001 versus empty or familiar group).

**Table 1 molecules-28-07300-t001:** Human monoamine oxidase (*h*MAO)-A and -B and lipid peroxidation-inhibitory activities of icariin and its metabolites.

Compounds	IC_50_ (μM) ^1^		
	*h*MAO-A	*h*MAO-B	Lipid Peroxidation
Icariin	421.74 ± 7.07	>500	301.03 ± 2.16
Icariside II	>500	>500	70.38 ± 2.04
Icaritin	>500	>500	81.33 ± 3.09
R-(–)-deprenyl HCl ^1^	17.41 ± 0.75	0.23 ± 0.02	ND
Trolox ^2^	ND	ND	32.33 ± 3.21

^1^ 50% inhibitory concentration (IC_50_) values (μM) were calculated from a dose-inhibition curve and expressed as mean ± standard deviation of triplicate experiments. ^2^ Positive controls for each assay. ND: Not detected.

## Data Availability

Not applicable.

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
