# Peer review of "Effects of Icariin and Its Metabolites on GPCR Regulation and MK-801-Induced Schizophrenia-Like Behaviors in Mice"

_molecules, 2023, doi:10.3390/molecules28217300_

Round 1

Reviewer 1 Report

Comments and Suggestions for Authors

In the manuscript of "Effects of Icariin and Its Metabolites on GPCR Regulation and 2

MK-801-Induced Schizophrenia-like Behaviors in Mice", The study explores the effects of icariin and its metabolites on GPCR regulation and their potential in mitigating schizophrenia-like behaviors induced by MK-801. This is a novel and original contribution to the field of neuropharmacology. The manuscript is very interesting and well prepared. However, there are certain suggestions to improve it for the better understanding.

1.      While the study effectively identifies the receptors targeted by icariin and its metabolites, it could benefit from a deeper exploration of the underlying molecular mechanisms that lead to these effects. The author may propose a possible mechanism and discuss it discussion section.

2.      The study reports half-maximal effective concentrations for icariin and icariside II, but more detailed dose-response relationships for all compounds would provide a more comprehensive understanding of their pharmacological effects.

3.      Consider including additional data, such as dose-response curves, to provide a more comprehensive view of the compounds' pharmacological effects.

4.      Consider discussing the potential clinical implications of the findings, including the relevance of targeting D3R and M2R for the treatment of cognitive and psychiatric disorders.

5.      While the conclusion provides a solid overview of the study's findings, it could benefit from a discussion on potential mechanisms underlying the observed effects. This could involve further molecular and cellular investigations.

6.      Consider incorporating a section in the conclusion that delves into the potential molecular and cellular mechanisms by which icariin and its metabolites exert their effects on D3R, M2R, and the neuroprotective pathways.

7.      Further elaborate on the potential clinical applications of icariin and its metabolites, and discuss the relevance of targeting D3R and M2R for the treatment of cognitive and psychiatric disorders.

8.      The conclusion strengthens the overall impact of the article by summarizing the key findings and highlighting the potential therapeutic implications of the study. By providing further mechanistic insights and discussing clinical applications, the article can further enhance its significance in the field.

 9.      I recommend it for revision before proceeding for publication. 

Author Response

Response to reviewer 1

In the manuscript of "Effects of Icariin and Its Metabolites on GPCR Regulation and 2

MK-801-Induced Schizophrenia-like Behaviors in Mice", The study explores the effects of icariin and its metabolites on GPCR regulation and their potential in mitigating schizophrenia-like behaviors induced by MK-801. This is a novel and original contribution to the field of neuropharmacology. The manuscript is very interesting and well prepared. However, there are certain suggestions to improve it for the better understanding.

  1. While the study effectively identifies the receptors targeted by icariin and its metabolites, it could benefit from a deeper exploration of the underlying molecular mechanisms that lead to these effects. The author may propose a possible mechanism and discuss it discussion section.

- Answer 1: Following the suggestion, we have proposed a possible mechanism and discussed it in the discussion section. Please check the revised manuscript.

(Line 273)

These in vitro and in silico results propose that icariside II could enhance dopaminergic function by acting as a D3R agonist that not only decreases cAMP formation but also regulates the Akt/glycogen synthase kinase-3β (GSK-3β) signaling pathway [32]. This activation of dopaminergic signaling pathway by D3R agonist action can effectively prevent motor symptoms of PD but has caused side effects related to cognitive function. However, recent studies have reported that multi-targeted ligands, such as SK609 targeting D3R and NET, can reduce these side effects [5].

(Line 312)

As a result, this study showed that icaritin has the potential to be employed as a medication to improve cognitive performance by functioning as an M2R antagonist that can control Ach release.

  1. The study reports half-maximal effective concentrations for icariin and icariside II, but more detailed dose-response relationships for all compounds would provide a more comprehensive understanding of their pharmacological effects.

- Answer 2: As described in result section (2.2. GPCR Target Screening of Icariin and Its Metabolites), we have added more detailed dose-response relationships for all compounds. Please check the revised manuscript.

(Line 98)

Thus, it was discovered that glycosides with a sugar linked to the C3- or C7-position are more efficient than icaritin itself in the D3R agonist action.

(Line 110)

Unfortunately, icariin and its metabolites did not exhibit any agonist/antagonist activity against dopamine D4 receptor (D4R), serotonin 1A receptor (5HT1AR), and calcitonin gene-related peptide receptor (CGRP) at a concentration of 100 μM.

  1. Consider including additional data, such as dose-response curves, to provide a more comprehensive view of the compounds' pharmacological effects.

- Answer 3: As shown in Figures 2b and 3a, we already provided the dose-response curves for D3R agonist effect of icariside II and D3R binding affinity of icariside II. Please check the manuscript.

  1. Consider discussing the potential clinical implications of the findings, including the relevance of targeting D3R and M2R for the treatment of cognitive and psychiatric disorders.

- Answer 4: Following the suggestion, we have included the potential clinical implications of the findings in the discussion section. Please check the revised manuscript.

(Line 273)

These in vitro and in silico results propose that icariside II could enhance dopaminergic function by acting as a D3R agonist that not only decreases cAMP formation but also regulates the Akt/glycogen synthase kinase-3β (GSK-3β) signaling pathway [32]. This activation of dopaminergic signaling pathway by D3R agonist action can effectively prevent motor symptoms of PD but has caused side effects related to cognitive function. However, recent studies have reported that multi-targeted ligands, such as SK609 targeting D3R and NET, can reduce these side effects [5].

In this study, icariside II not only showed the D3R agonist effect but also showed a V1AR antagonist effect. Vasopressin (AVP), a neuropeptide with three receptors: V1A, V1B, and V2, regulates amygdaloid activity in response to threatening stimuli as well as anxious mood. Since V1AR-expressing regions modulate anxiety and fear, studies on rodents have mainly examined how V1AR affects anxious behavior [33]. Recently, anthraquinone derived from Cassia seed, which is dual D3R/V1AR agonist, significantly alleviated memory impairment and neuronal damage in the transient forebrain ischemia animal model [34]. Taken together, it is believed that icariside II could improve motor symptoms seen in neurodegenerative disorders through D3R agonist action and alleviate anxious behavior and memory performance through the V1AR antagonist action. However, additional mechanism studies and in vivo validation should be conducted.

  1. While the conclusion provides a solid overview of the study's findings, it could benefit from a discussion on potential mechanisms underlying the observed effects. This could involve further molecular and cellular investigations.

- Answer 5: Following the suggestion, we have included the potential mechanisms underlying the observed effects in the conclusion section. Please check the revised manuscript.

(Line 485)

Icariside II could enhance the dopaminergic function by acting as a selective D3R agonist that not only inhibits cAMP formation but also regulates the Akt/GSK-3β signaling pathway. In addition, icariside II could alleviate anxious behavior and memory performance through the V1AR antagonist action. Furthermore, icaritin could increase the availability of acetylcholine by acting as an M2R antagonist, thereby improving learning and memory performance.

  1. Consider incorporating a section in the conclusion that delves into the potential molecular and cellular mechanisms by which icariin and its metabolites exert their effects on D3R, M2R, and the neuroprotective pathways.

- Answer 6: As in reply to comment 1, we have included the potential mechanisms underlying the observed effects in the conclusion section. Please check the revised manuscript.

  1. Further elaborate on the potential clinical applications of icariin and its metabolites, and discuss the relevance of targeting D3R and M2R for the treatment of cognitive and psychiatric disorders.

- Answer 7: In the discussion section, we have described the metabolism of icariin by human intestinal bacteria as well as BBB permeability potential, which is most important for CNS drug development. Please check the manuscript. In addition, as in reply to comments 1 and 4, we have added the more detailed description of the possible mechanisms involved in the protective effects of icariin and its metabolites against cognitive and neurodegenerative disorders. Please check the revised manuscript.

  1. The conclusion strengthens the overall impact of the article by summarizing the key findings and highlighting the potential therapeutic implications of the study. By providing further mechanistic insights and discussing clinical applications, the article can further enhance its significance in the field.

- Answer 8: As in reply to comment 6, we have included the potential mechanisms underlying the observed effects in the conclusion section. Please check the revised manuscript.

  1. I recommend it for revision before proceeding for publication. 

- Answer 9: Thank you for your valuable comments on our manuscript. We have revised the manuscript point by point to the reviewer’s comments. Please check the revised manuscript.

Reviewer 2 Report

Comments and Suggestions for Authors

The manuscript describes “Effects of Icariin and Its Metabolites on GPCR Regulation and MK-801-Induced Schizophrenia-like Behaviors in Mice.  The title of the manuscript is wider than the subject incorporated in it. The author wants to study the regulatory effects of icariin and its active metabolites (icariside II and icaritin) on certain G-protein-coupled receptor targets, particularly their association with neuronal disorders.The manuscript is well written, which I overall accept. The parameters chosen for strategy building are traditional and informative. But there are certain areas that need more clarification.

Abstract: “Icariin and icariside II exhibited selective agonist activity towards the dopamine D3 receptor (D3R), 21with respective half-maximal effective concentrations of >100 μM and 13.29 μM.” This sentence is quite confusing please reframe it.

Introduction: To enhance it, essential points should be highlighted. Please extend this section to provide a more comprehensive understanding of the subject and include the latest references.

Materials and methods: The structure and explanation of Section 4.2 can be improved. What is the purity of the E. koreanum? Please provide all the information, unless you have already given an appropriate reference.

Section 4.7: Could you provide more information about the parameters and criteria used for selecting the most populated cluster and the ultimate docking outcome, including any specific cutoff values for binding energy?

Results: Section 2.6 and 2.7 can be improved with more details since there is enough to explain.

Discussion part is very weak need to the reorganized because few parts of explanations are not connected and missing with latest references.

General comments:

Please double-check the manuscript for errors. The content presented here is more theoretical. There are many places that need to be updated with the latest reference. I suggest the authors to go for English corrections with a native speaker or a professional company.

If possible, please provide the graphical abstract. This can help readers to understand better.

Conclusion:

The manuscript is interesting but strategic connectivity to the proposed hypothesis is lacking in certain places. Since I encountered some difficulty in reading and comprehending certain sections of the manuscript, the article requires critical revisions and the inclusion of specific details. I believe that the manuscript addresses important issues, employs interesting approaches and techniques, and can contribute to understanding the regulatory role of icariin and its active metabolites in neuronal disorders." Therefore, I consider this manuscript suitable for publication after minor revision in Molecules.

Author Response

V

Response to reviewer 2

The manuscript describes “Effects of Icariin and Its Metabolites on GPCR Regulation and MK-801-Induced Schizophrenia-like Behaviors in Mice”.  The title of the manuscript is wider than the subject incorporated in it. The author wants to study the regulatory effects of icariin and its active metabolites (icariside II and icaritin) on certain G-protein-coupled receptor targets, particularly their association with neuronal disorders.The manuscript is well written, which I overall accept. The parameters chosen for strategy building are traditional and informative. But there are certain areas that need more clarification.

  1. Abstract: “Icariin and icariside II exhibited selective agonist activity towards the dopamine D3 receptor (D3R), 21with respective half-maximal effective concentrations of >100 μM and 13.29 μM.” This sentence is quite confusing please reframe it.

- Answer 1: Following the suggestion, we have revised the abstract. Please check the revised manuscript.

(Line 21)

Icariin and icariside II exhibited selective agonist activity towards the dopamine D3 receptor (D3R), with respective half-maximal effective concentrations of >100 μM and 13.29 μM.

→ Icariside II exhibited selective agonist activity towards the dopamine D3 receptor (D3R), with half-maximal effective concentrations of 13.29 μM.

  1. Introduction: To enhance it, essential points should be highlighted. Please extend this section to provide a more comprehensive understanding of the subject and include the latest references.

- Answer 2: Following the suggestion, we have added more information regarding GPCR drugs based on the latest references. Please check the revised manuscript.

  1. Materials and methods: The structure and explanation of Section 4.2 can be improved. What is the purity of the E. koreanum? Please provide all the information, unless you have already given an appropriate reference.

- Answer 3: Epimedium koreanum is plant source and we provided the ethanol extract of Epimedium koreanum from KIST natural product library.

  1. Section 4.7: Could you provide more information about the parameters and criteria used for selecting the most populated cluster and the ultimate docking outcome, including any specific cutoff values for binding energy?

- Answer 4: Following the suggestion, we have added more information about the parameters for molecular docking analysis. Please check the revised manuscript.

(Line 405)

A grid box sized 80 × 90 × 80 with a spacing of 0.375 Å was used to cover the favorable target binding sites. The grid centers were 0.86, -15.073, and 9.582 for 3pbl and 6.663, 0.524, and -2.976 for 3uon.

  1. Results: Section 2.6 and 2.7 can be improved with more details since there is enough to explain.

- Answer 5: Following the suggestion, we have improved the result section for in vivo experiments. Please check the revised manuscript.

(Line 187 for section 2.6.)

A prepulse inhibition (PPI) test is a widely used paradigm for assessing sensorimotor gating in humans and rodents. The PPI of the startle response (SR) occurs when a mild prepulse stimulus is provided 30-500 ms before the startling stimulus. Patients with psychiatric diseases like schizophrenia have been observed to have reduced PPI, which is assumed to represent sensorimotor gating failure [16].

(Line 210 for section 2.7.)

The primary principle of this experiment is based on a subject mouse’s free choice to spend time in any of three box compartments throughout two experimental sessions, involving indirect interaction with one or two unfamiliar mice [17].

  1. Discussion part is very weak need to the reorganized because few parts of explanations are not connected and missing with latest references.

- Answer 6: Following the suggestion, we have added more explanations in the discussion section with the latest references. Please check the revised manuscript.

  1. General comments:

Please double-check the manuscript for errors. The content presented here is more theoretical. There are many places that need to be updated with the latest reference. I suggest the authors to go for English corrections with a native speaker or a professional company. If possible, please provide the graphical abstract. This can help readers to understand better.

- Answer 7: Following the suggestion, we have updated the references and provided the graphical abstract. And our manuscript was proofread in English (please check the editing certificate).

  1. Conclusion:

The manuscript is interesting but strategic connectivity to the proposed hypothesis is lacking in certain places. Since I encountered some difficulty in reading and comprehending certain sections of the manuscript, the article requires critical revisions and the inclusion of specific details. I believe that the manuscript addresses important issues, employs interesting approaches and techniques, and can contribute to understanding the regulatory role of icariin and its active metabolites in neuronal disorders." Therefore, I consider this manuscript suitable for publication after minor revision in Molecules.

- Answer 8: Thank you for your valuable comments on our manuscript. We have revised the manuscript point by point to the reviewer’s comments. Please check the revised manuscript.

Round 2

Reviewer 1 Report

Comments and Suggestions for Authors

The authors responded to the reviewer's comments and the manuscript is improved.